# Prospective observational study of peripheral intravenous cannula utilisation and frequency of intravenous fluid delivery in the emergency department—Convenience or necessity?

Michael Willis[1]*, Efrem Colonetti[2], Ali Bakir[3], Yousef Jamal Alame[4],
Megan Annetts[5], Deren T. Aygin[6], Amina Daou[7], Sultan Farooq[8], Nicholas A. Fine[1],
Gozde Firat[9], Benjamin Goozee[10], Anuj Neelesh Gupta[11], Charlotte Hubbett[1], Nicole Shun
Yee Loi[12], Laura Maciejec-Biskup[13], Merline Gabriela Muthukumar[14], Jason Pott[1],
Benjamin M. Bloom[1], Maria Lorenza Muiesan[2], Tim Harris[1]*

1 Barts Health NHS Trust, London, England, 2 Department of Clinical and Experimental Sciences University
of Brescia, Brescia, Italy, 3 The Ivers Practice, Iver, England, 4 Hampshire Hospital NHS Foundation Trust,
Hampshire, England, 5 Monash Children's Hospital, Melbourne, Australia, 6 Mid and South Essex NHS
Foundation Trust, Chelmsford, England, 7 Guy's and St Thomas' NHS Foundation Trust, London, England,
8 Whittington Health NHS Trust, London, England, 9 The Royal Free London NHS Foundation Trust,
London, England, 10 St John of God Subiaco Hospital, Perth, Australia, 11 Dorset County Hospital NHS
Foundation Trust, Dorchester, United Kingdom, 12 The Prince Charles Hospital, Brisbane, Australia,
13 Royal North Shore Hospital, Sydney, Australia, 14 Lewisham and Greenwich NHS Foundation Trust,
London, England

* m.willis@doctors.org.uk (MW); t.harris@qmul.ac.uk (TH)

pone.0305276

School of Medicine, UNITED STATES

## Abstract

### Background

Peripheral Intravenous Cannulas (PIVCs) are frequently utilised in the Emergency Department (ED) for delivery of medication and phlebotomy. They are associated with complications and have an associated cost to departmental resources. A growing body of international research suggests many of the PIVCs inserted in the ED are unnecessary.

### Methods

The objective of this study was to determine the rates of PIVC insertion and use. This was a prospective observational study conducted in one UK ED and one Italian ED. Adult ED patients with non-immediate triage categories were included over a period of three weeks in the UK ED in August 2016 and two weeks in the Italian ED in March and August 2017. Episodes of PIVC insertion and data on PIVC utilisation in adults were recorded. PIVC use was classified as necessary, unnecessary or unused. The proportion of unnecessary and unused PIVCs was calculated. PIVCs were defined as unnecessary if they were either used for phlebotomy only, or solely for IV fluids in patients that could have potentially been hydrated orally (determined against *a priori* defined criteria). PIVC classified as unused were not used for any purpose.

**Data Availability Statement:** All relevant data are within the manuscript and its Supporting information files.

**Funding:** The authors received no specific funding for this work.

**Competing interests:** The authors have declared that no competing interests exist.

## Results

A total of 1,618 patients were included amongst which 977 PIVCs were inserted. Of the 977 PIVCs, 413 (42%) were necessary, 536 (55%) were unnecessary, and 28 (3%) were unused. Of the unnecessary PIVCs, 473 (48%) were used solely for phlebotomy and 63 (6%) were used for IV fluids in patients that could drink.

## Conclusions

More than half of PIVCs placed in the ED were unnecessary in this study. This suggests that clinical decision making about the benefits and risks of PIVC insertion is not being performed on an individual basis.

## Introduction

Over one billion peripheral intravenous cannulas (PIVCs) are inserted worldwide each year [1], with an estimated 60% of hospitalised patients receiving PIVCs over the course of their admission [2]. PIVCs are frequently inserted in the Emergency Department (ED) for blood sample collection and are most commonly used for the delivery of IV fluid [3, 4]. Previous studies have not assessed if IV fluid delivery followed best practice and/or was required for the clinical situation. Insertion of PIVCs may alter clinician's prescribing behaviour potentially influencing the prescribing of IV fluids where oral hydration could be used [3]. Data suggests many ED patients can be hydrated orally and supplementary IV fluids are not always required [5, 6]. Identifying the incidence of unnecessary IV fluid delivery could assist in guiding policy concerning the insertion of and appropriate usage of PIVCs.

Previous studies have shown rates of unused or unnecessary ED PIVCs ranging from 9% to 83% [3–11]. However, most of these studies used retrospective case-note reviews to determine the insertion rate and usage of PIVCs.

PIVCs carry an associated cost and risk, most notably for catheter-associated infection and *Staphylococcus aureus* bacteraemia, with an estimated 10,028 annual cases in the US [12] and other studies in inpatients on medical, surgical and ICUs (Intensive Care Units) suggest a rate of between 0.03% to 0.1% [7, 13]. This risk of catheter-associated infection has been reported to be significantly greater when inserted in the ED compared to inpatient wards (odds ratio [OR] 6.0, p<0.001) [12]. Other risks associated with the procedure include pain, phlebitis, occlusion and medication extravasation [14, 15]. The cost associated with inserting PIVCs in two Australian studies ranges from AUD$16.40 to 22.79 (USD$11.84–16.45, GBP£8.87–12.32 at time of publication) [11, 16]. There is also an significant environmental impact from non-recyclable plastic packaging and materials required to insert and dress an unnecessary/unused PIVC.

We therefore performed a prospective observational study to determine the rates of PIVC insertion and describe their use in the ED in two hospitals; one in the UK ED and one in Italy. Secondly, we sought to assess the proportion of patients receiving IV fluid who could have potentially been hydrated orally against *a priori* defined criteria.

## Methods

### Study setting and design

This study took place in the EDs of two urban, tertiary centres; centre A in UK and centre B in Italy. Centre A is a major trauma centre, teaching hospital and stroke centre seeing over

135,000 patients per year in the ED. Centre B is also a trauma centre and teaching hospital see-ing over 80,000 patient per year in the ED. In centre A, data were collected prospectively by observed PIVC utilisation in patients over 3 weeks. Data was collected between 0800–2200 for 21 consecutive days in August 2016. In centre B data was prospectively collected in random 6-hour blocks between 0800–2000 in two 7-day periods in March and August 2017. The times for data collection were determined by researcher availability.

The study was reviewed by the research and audit departments for each institution and assessed via the institutional research assessment system, both of which delegated the study as not requiring written informed consent. There were no interventions, no patient identifying data was collected, no deviation from usual care occurred and no direct contact between the study team and patients.

## Selection of participants

The inclusion criteria was all adult patients >18 years who were triaged to a non-resuscitation area of the ED and were discharged/admitted prior to the end of the data collection period. Patients were excluded if a PIVC had been inserted before ED arrival, if transferred to resusci-tation areas, if presenting with minor injuries or if patients were still in the ED at the end of the study team's data collection period (as it was not possible to prospectively determine PIVC use).

## Methods of measurement

We *a priori* defined 'unnecessary' PIVCs as PIVCS utilised for phlebotomy only or used for IV fluids administration where none of the *a priori* criteria for IV fluids were met and no other IV treatment was given. "Unused" PIVCs were those never utilised for phlebotomy, nor IV fluid/medication delivery. We have combined these both of these definitions as "unused and unnec-essary" PIVCs. Necessary PIVCs were defined as used for IV medications or IV fluids meeting the *a priori* defined criteria for IV fluid administration as below.

The data collection team received training concerning the study objectives, definitions, and an electronic tablet with an online database for collecting data. Data concerning PIVC inser-tion and utilisation were prospectively recorded from patient arrival until discharge from the ED through direct observation, prospective ED record review and/or clarification with the patient's clinician. Data collected included: arrival date and time, biological sex, age, arrival mode, PIVC insertion location, PIVC gauge, grade of staff inserting the PIVC, PIVC usage (plus date and time for each use), *a priori* assessment of indication for intravenous fluid treat-ment and ED discharge time. All data were anonymised at the point of data entry.

The study team assessed the indication for administering IV fluids against a set of *a priori* defined clinical indications based on National Institute for Health and Care Excellence (NICE, UK) guidance for intravenous therapy in adults [17]: emesis/nausea, IV fluid required to give medications/blood products/contrast media, fluid resuscitation required to improve oxygen delivery and organ perfusion, correction of electrolyte abnormalities, and/or the patient being held nil by mouth (for a proposed procedure or as GCS < 15 or swallowing impairment/await-ing swallowing assessment). Where there was no clear indication for prescribing IV fluids this was classified as 'unnecessary'.

## Primary data analysis

All data is presented using frequency and percentages. The results were analysed using Micro-soft Excel.

**Table 1. Demographic data on patients receiving a PIVC.**

|  | Centre A | Centre B |
|---|---|---|
| **Average age** | 49 | 63 |
| **Gender** | | |
| Male | 48% | 54% |
| Female | 52% | 46% |
| **PIVC gauge** | | |
| 14G/Orange | 1% | 0.25% |
| 16G/Grey | 2% | |
| 17G/White | | 9.5% |
| 18G/Green | 14% | 67% |
| 20G/Pink | 68% | 22% |
| 22G/Blue | 12% | 0.25% |
| 24G/Yellow | 1% | |
| 26G/Violet | 1% | |
| **Staff grade inserting PIVC** | | |
| Doctor | 4% | 1% |
| Emergency Department Assistants | 33% | |
| Medical student | <1% | |
| Nurse | 62% | 98% |
| Other | <1% | 0.75% |

Demographics included insertion site, gauge and grade of staff inserting the PIVC. Emergency Department Assistants (EDAs) are trained technicians aiding in phlebotomy, observations, and ECGs. Percentages may not add to 100 given rounding.

## Results

Baseline demographic data is presented in Table 1. Data were collected on 1618 patients, of which 977 had a PIVC inserted. Of the 977 PIVCs, 413 (42%) were necessary, 536 (55%) were unnecessary, and 28 (3%) were unused. Of the unnecessary PIVCs, 473 (48%) were used solely for phlebotomy and 63 (6%) were used for administration of IV fluids in patients that could drink. (Fig 1, Table 2). 130 (13%) PIVCs were utilised for IV fluids only, of which 63 (6%) did not meet the *a priori* criteria for IV fluid administration and were therefore classified as 'unnecessary'.

## Discussion

This is one of the largest international prospective studies on PIVC use published to date. Data were collected on 1,618 patients with 977 (60%) having a PIVC placed. 564 (58%) of the PIVCs were classified as unnecessary, or unused. We defined PIVC placement as 'unnecessary' where PIVCs were utilised for phlebotomy only (473 (48%)), or for IV fluids administration where no *a priori* criteria were met and no other IV medication was given (63 (6%)). The number of unused PIVCs in the study was 28 (3%).

 Centre B had a slightly older demographic with both centres having similar male:female ratios. Centre A had a higher number of PIVCs inserted, while for both sites the majority of PIVC insertion was by a nurse, EDA, and < 5% by physicians. Centre B inserted larger PIVCs on average. Both centres had similarly (high) numbers of unnecessary and unused PIVCs of 56% and 60% respectively, rates midway between previous studies, where PIVC utilisation

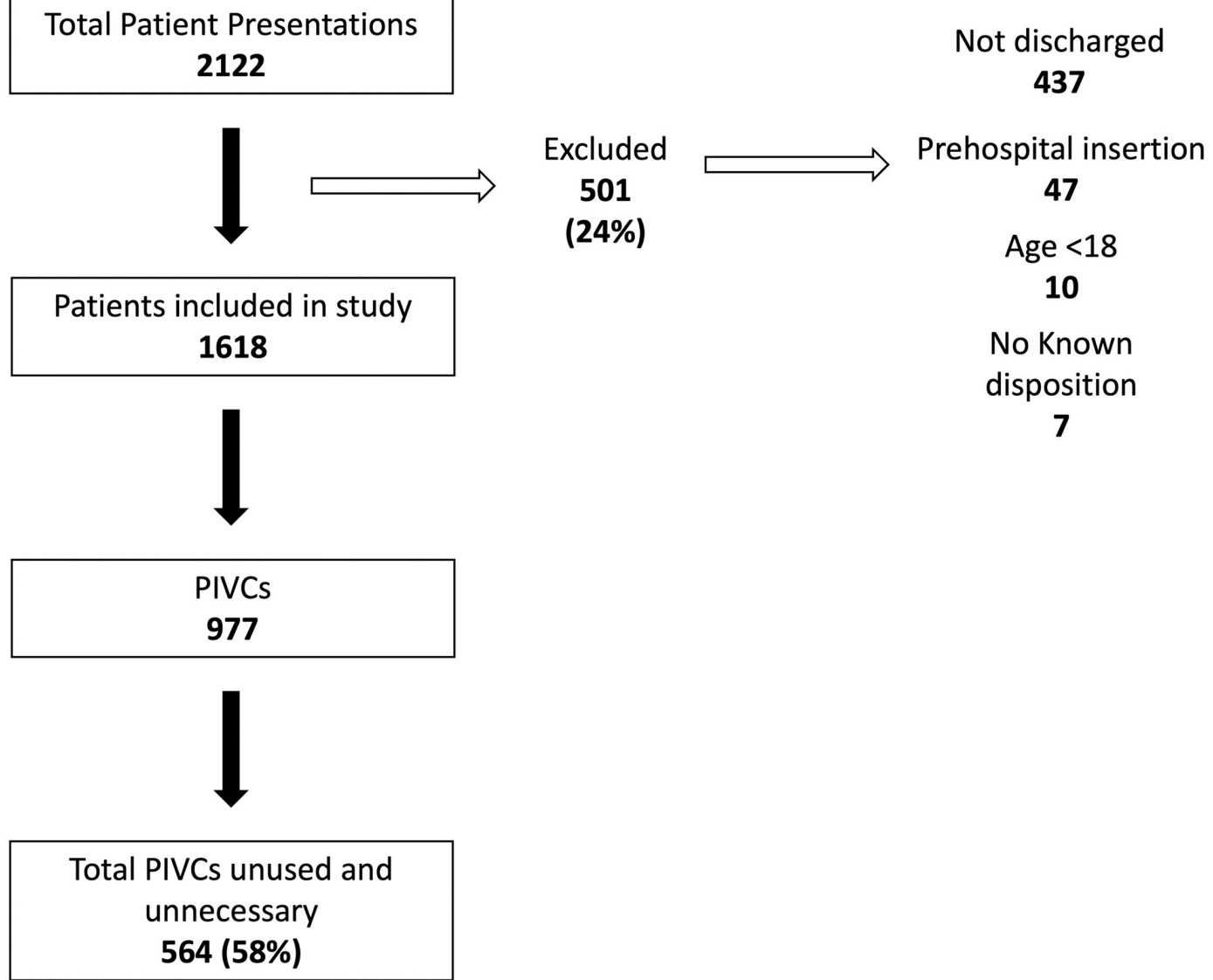

**Fig 1. Flow diagram of PIVCs included in the study.** Peripheral intravenous catheters inserted and determined to be unused and unnecessary. Data combined from centre A (UK) and centre B (Italy).

rates between 35–72% [3, 6, 8–10, 18]. However, most of these studies are retrospective case note reviews. Fry et al. [4] reported a significantly lower rate than reported in this study with 9% of PIVCs classified by the authors as unnecessary and unused [4]. This was a retrospective study and the authors cited poor documentation of PIVC insertion site, size and use. The authors also did not account for inappropriate IV fluid usage which may have increased the number of PIVCs identified as unnecessary. Applying the criteria for unnecessary cannulation used in this study, 23% of PIVCs would be classified as unnecessary—with 30% PIVCs being used solely for phlebotomy. Limm et al. [3] reported an ED PIVC insertion rate of 15% with 50% unused up to 72 hours post-insertion in the ED. In a prospective study, Abbas et al. reported 86/106 (81%) patients admitted from the ED as having PIVCs placed, with 42 (49%) unused for the duration of the PIVC with 66% remaining in situ for >72 hours [19]. This

**Table 2. Summary of PIVC use for patients presenting to centre A and B with combined data.**

|  | Combined | Centre A | Centre B |
|---|---|---|---|
| PIVCs inserted (% of total patients) | 977 (60%) | 578 (50%) | 399 (87%) |
| Blood sampling | 874 (89%) | 492 (85%) | 382 (96%) |
| IV fluids | 268 (27%) | 215 (37%) | 53 (13%) |
| Analgesia | 130 (13%) | 41 (7%) | 89 (22%) |
| Non-analgesic medication | 245 (25%) | 177 (31%) | 68 (17%) |
| Radiological contrast | 12 (1%) | 5 (1%) | 7 (2%) |
| Blood transfusions | 4 (0%) | 2 (0%) | 2 (1%) |
| Necessary PIVCs | 413 (42%) | 255 (44%) | 158 (40%) |
| Unnecessary PIVCs |  |  |  |
| No *a priori* criteria met for IV fluid delivery | 63 (6%) | 48 (8%) | 15 (4%) |
| Phlebotomy only | 473 (48%) | 248 (43%) | 225 (56%) |
| Unused PIVCs | 28 (3%) | 27 (5%) | 1 (<1%) |
| Unused and unnecessary | 564 (58%) | 323 (56%) | 241 (60%) |

We defined 'unnecessary' PIVCs as PIVCS utilised for phlebotomy only or used for IV fluids administration where none of the *a priori* criteria were met and no other treatment was given. Unused PIVCs were those never utilised for IV fluids/medications or phlebotomy. Necessary PIVCs were defined as used for IV medications or IV fluids meeting the a priori defined criteria for IV fluid administration.

Percentages reflect proportion of total cannulas where otherwise specified. Percentages may sum to more than 100 due to individual PIVCs being used for multiple therapies.

small study collected data over 2 weeks describing similar data to the work reported in this journal.

PIVC insertion may alter clinician behaviour to prescribe IV fluids where patients could hydrate orally or where there is no therapeutic indication for IV fluids. To investigate this *a priori* criteria were defined for IV fluid delivery. The National Institute for Health and Care Excellence (NICE, UK) recommends patients not requiring fluid resuscitation (after initial assessment) receive a trial of oral or enteral fluid hydration before commencing IV fluids [17]. IV fluids are frequently prescribed for conditions where there is evidence against their efficacy, such as renal colic [20] and alcohol intoxication [21]. In this study IV fluids were administered for 130 (13%) of patients. In 48% of these patients, the *a priori* criteria defining clear patient benefit for IV fluid therapy were not met. This may reflect poor knowledge of national guidelines, lack of available oral hydration within the EDs and/or clinician belief in a therapeutic benefit for IV fluids. A recent large, prospective, before and after study demonstrated a PIVC usage rate of 70.5% pre-intervention and 83% post-intervention [11]. The authors defined a PIVC as unused if it was not utilised for IV fluids, medications or blood products within 24 hours and similarly did not distinguish those IV fluids with clear patient benefit, which may have further increased their defined "unused" PIVCs. Gentges et al. in a retrospective patient chart review reported a PIVC insertion rate in the ED of 55% and a significantly lower rate of unused PIVCs at 16% [5]. There was no definition for "unnecessary" IV fluid administration and the authors suggested that oral rehydration protocols were not well known; potentially increasing the rate of unnecessary IV fluid prescribing. Guihard et al. [22] reported 43% of PIVCs as unnecessary, with the administration of IV medications where an equally effective oral medication was available, most commonly analgesia (77% of cases) [22].

We report only on the use of PIVCs in the ED, some PIVCs not used in the ED may have been used on admitted patients. Guihard et al. [22] prospectively monitored all PIVCs inserted

in the ED for 1 week and followed those admitted to medical wards. They counted 'inappropriate' PIVCs using similar *a priori* criteria as the study presented here and reported 43% of PIVCs to be unused or unnecessary. Lederle et al. [9] prospectively monitored 484 inpatients with PIVCs over a 6-week period and reported 33% had >2 consecutive days with no PIVC use and no reason for maintaining IV access. Data suggests that the longer a PIVC remains in situ the greater the risk of thrombophlebitis, infection and bacteraemia [1]. A recent prospective cohort study reported a significant increase in the incidence of bacteraemia following the introduction of a clinically indicated PIVC replacement strategy, which led to longer indwelling times [23]. Inserting PIVCs based on clinical need at the time they are required may reduce the total time for in situ cannula placement and associated risks.

Eleven (<1%) patients in centre A required transfer to the resuscitation area consequent upon clinical deterioration and in 7 of these patients the PIVC was unused until transfer. In a letter Lewindon highlighted the benefits of prophylactically placing PIVCs in the ED, where practitioners may be more familiar with the procedure, arguing PIVC placement may potentially prevent adverse outcomes if patients were to deteriorate and the difficulties surrounding urgent cannulation [24]. Other authors have stressed there may be a culture of convenience, and more junior members of staff view PIVC placement as part of the admissions process [25].

PIVC placement is regarded as painful by many patients, warranting analgesia [26, 27]. It has been shown to be more expensive [24] and take more time than phlebotomy alone [10] and utilises more resources and single-use plastics. In the study reported here 874 (89%) PIVCs were used for blood sampling; including 473 (48%) used only for phlebotomy with no IV medication or IV fluid delivery. Phlebotomy would be a less expensive and less painful alternative to PIVC insertion.

## Limitations and recommendations for future work

There are several limitations to the study. Data was collected in two large tertiary inner-city EDs and the findings may not be generalisable to other countries or non-tertiary units with differing levels of trainee medical staff. We did not include patients triaged to resuscitation areas where a higher clinical risk suggests PIVC placement is more likely to be required. The study population did not include children or adolescents (<18 years) where different clinical behaviour may be observed based on both clinician and patient expectations [8].

Assessing the appropriateness of IV fluids is difficult to objectify with different clinicians offering different perspectives. There is data to suggest that clinicians do prescribe IV fluids for patients with no clear clinical benefit [20, 21]. There are no published guidelines for prescribing intravenous fluids across the wide spectrum of ED patients. The criteria to define IV fluid as 'unnecessary' used in this study are unvalidated and were developed for this study.

Our study was limited by data collection team availability and did not assess patients attending between 2200 and 0800 at centre A, and before/after the 6-hour random blocks at centre B. This convenience sample may not be a true reflection of patients presenting at other times. We excluded patients' data from analysis if they were not discharged or admitted to the hospital by the end of a data collection period, as a result data on 501 patients were excluded. (Fig 1).

## Conclusion

This study provides further evidence that a high proportion of PIVCs placed in the ED are not used to deliver therapy. 977/1618 (60%) of study patients had a PIVC inserted with 564 (58%) identified as unnecessary or unused. In 130 (13%) patients, the PIVCs were used only for IV fluids (with or without blood samples) with 63 (6%) not meeting *a priori* criteria for IV fluid

therapy, where oral as opposed to IV fluids could have been prescribed. 473 (48%) PIVCs were used for blood draw only. ED patients are by their nature undifferentiated and at risk of clinical deterioration and may require urgent, unexpected administration of IV medication which may explain clinicians having a low threshold to place PIVCs. In the study reported here, <1% of patients required transfer to the resuscitation area for urgent treatment. Reducing the rate of PIVC placement may reduce costs, complications, patients' pain and the rate of unnecessary prescriptions of IV fluids and medications.

## Supporting information

**S1 File.**
(DOCX)

## Author Contributions

**Conceptualization:** Michael Willis, Efrem Colonetti, Ali Bakir, Yousef Jamal Alame, Megan Annetts, Deren T. Aygin, Amina Daou, Sultan Farooq, Nicholas A. Fine, Gozde Firat, Benjamin Goozee, Anuj Neelesh Gupta, Charlotte Hubbett, Nicole Shun Yee Loi, Laura Maciejec-Biskup, Merline Gabriela Muthukumar, Jason Pott, Benjamin M. Bloom, Maria Lorenza Muiesan, Tim Harris.

**Data curation:** Michael Willis, Efrem Colonetti, Ali Bakir.

**Formal analysis:** Michael Willis, Efrem Colonetti, Ali Bakir, Benjamin M. Bloom.

**Investigation:** Michael Willis, Efrem Colonetti, Ali Bakir, Yousef Jamal Alame, Megan Annetts, Deren T. Aygin, Amina Daou, Sultan Farooq, Nicholas A. Fine, Gozde Firat, Benjamin Goozee, Anuj Neelesh Gupta, Charlotte Hubbett, Nicole Shun Yee Loi, Laura Maciejec-Biskup, Merline Gabriela Muthukumar, Benjamin M. Bloom, Tim Harris.

**Methodology:** Michael Willis, Efrem Colonetti, Ali Bakir, Yousef Jamal Alame, Megan Annetts, Deren T. Aygin, Amina Daou, Sultan Farooq, Nicholas A. Fine, Gozde Firat, Benjamin Goozee, Anuj Neelesh Gupta, Charlotte Hubbett, Nicole Shun Yee Loi, Laura Maciejec-Biskup, Merline Gabriela Muthukumar, Jason Pott, Benjamin M. Bloom, Maria Lorenza Muiesan, Tim Harris.

**Project administration:** Michael Willis, Efrem Colonetti, Ali Bakir, Tim Harris.

**Supervision:** Benjamin M. Bloom, Maria Lorenza Muiesan, Tim Harris.

**Visualization:** Michael Willis, Efrem Colonetti.

**Writing – original draft:** Michael Willis, Tim Harris.

**Writing – review & editing:** Michael Willis, Efrem Colonetti, Ali Bakir, Yousef Jamal Alame, Megan Annetts, Deren T. Aygin, Amina Daou, Sultan Farooq, Nicholas A. Fine, Gozde Firat, Benjamin Goozee, Anuj Neelesh Gupta, Charlotte Hubbett, Nicole Shun Yee Loi, Laura Maciejec-Biskup, Merline Gabriela Muthukumar, Jason Pott, Benjamin M. Bloom, Maria Lorenza Muiesan, Tim Harris.

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
