## [Editor Report · Decision Letter 0]

2 Jul 2023

PONE-D-23-09905Prospective observational study of peripheral intravenous cannula utilisation and frequency of intravenous fluid delivery in the emergency department – convenience or necessity?PLOS ONE

Dear Dr. Willis,

Thank you for submitting your manuscript to PLOS ONE. After careful consideration, we feel that it has merit but does not fully meet PLOS ONE’s publication criteria as it currently stands. Therefore, we invite you to submit a revised version of the manuscript that addresses the points raised during the review process.

Dear Author,

Your manuscript titled "Prospective observational study of peripheral intravenous cannula utilisation and frequency of intravenous fluid delivery in the emergency department – convenience or necessity?" has gone through editorial review. In order meet journal requirements, the following minor corrections need to be made.

1. Data availability - Please state where (that is, in what database) or from whom the data analyzed for this study can be retrieved.

2. Insert a title for figure 1.

3. References - You used et al after listing only one author for most of your citations. With Vancouvre referencing the norm is to list six authors putting et al. Please review the references according.

Once all changes have been made reviewers will be invited.

Thank you.

We look forward to receiving your revised manuscript.

Kind regards,

Ochuwa Adiketu Babah, FWACS, FMCOG

Academic Editor

PLOS ONE
---

## [Author Response · Author response to Decision Letter 0]

21 Jul 2023

We thank you for your consideration and review of the original research article entitled “Prospective observational study of peripheral intravenous cannula utilisation and frequency of intravenous fluid delivery in the emergency department – convenience or necessity?” 

Please see below a response to the minor corrections requested:

1. Data availability - Please state where (that is, in what database) or from whom the data analyzed for this study can be retrieved.

Response: the data can be requested from myself (Dr Michael Willis), in a Microsoft Excel format.

2. Insert a title for figure 1.

Response: A title is present on lines 142 and 143 for figure 1. Please let me know if this needs to be rectified/in a different formatting.

3. References - You used et al after listing only one author for most of your citations. With Vancouvre referencing the norm is to list six authors putting et al. Please review the references according.

Response: this has been rectified in the references section

Thank you once again for your time in considering this work.

---

## [Decision Letter · Decision Letter 1]

20 Dec 2023

PONE-D-23-09905R1Prospective observational study of peripheral intravenous cannula utilisation and frequency of intravenous fluid delivery in the emergency department – convenience or necessity?PLOS ONE

Dear Dr. Willis,

Thank you for submitting your manuscript to PLOS ONE. After careful consideration, we feel that it has merit but does not fully meet PLOS ONE’s publication criteria as it currently stands. Therefore, we invite you to submit a revised version of the manuscript that addresses the points raised during the review process.

We look forward to receiving your revised manuscript.

Kind regards,

Amit Bahl, MD

Academic Editor

PLOS ONE

Reviewers' comments:

Reviewer's Responses to Questions

**Comments to the Author**

1. If the authors have adequately addressed your comments raised in a previous round of review and you feel that this manuscript is now acceptable for publication, you may indicate that here to bypass the “Comments to the Author” section, enter your conflict of interest statement in the “Confidential to Editor” section, and submit your "Accept" recommendation.

Reviewer #1: (No Response)

Reviewer #2: All comments have been addressed

2. Is the manuscript technically sound, and do the data support the conclusions?

Reviewer #1: Partly

Reviewer #2: Partly

3. Has the statistical analysis been performed appropriately and rigorously? 

Reviewer #1: N/A

Reviewer #2: I Don't Know

4. Have the authors made all data underlying the findings in their manuscript fully available?

Reviewer #1: Yes

Reviewer #2: Yes

5. Is the manuscript presented in an intelligible fashion and written in standard English?

Reviewer #1: No

Reviewer #2: Yes

6. Review Comments to the Author

Reviewer #1: Thanks to the authors for submitting this article that looks at PIVC used in the ED setting. This is a useful topic given the risks associated with PIVC and the environmental impact. I have a few queries for the authors to clarify areas in the article

Introduction

Suggest re-structuring the introduction- 4th paragraph can be integrated with the first, then follow with risks associated with PIVC and then discuss unused/unnecessary PIVC. Authors could also consider discussing the environmental impact of the inappropriate use of PIVC

Line 63: where were the other studies conducted?

Lines 63-65- the sentence related to OR is not quite clear: are you saying that the risk of catheter associated infections is 6 times greater when inserted in the ED compared to inpatient wards? What is the p value? I’m not sure what the OR of 2.3 is referring to

Please also include where this study was conducted and if it was a large or small study eg. Are these results from a single centre ED in X country?

What is the rational for focusing only on the use of PIVC for IV fluids? What about unnecessary use related to medication? Or generally on the appropriate use of PIVC?

The first part of the aim isn’t quite clear- are you assessing the prevalence of PIVC use in the ED or are you describing the indications for PIVC use in the ED?

Methods

Need to specifically say that the study site were ED centers

Suggest providing a brief description of the ED sites- size, volume of patients, staffing, training, types of patients

Selection of participants might be better presented as inclusion and exclusion criteria. Unlikely that all patients would have been followed given the design used. Rather the inclusion criteria would have been adult patients, presenting to the ED with X conditions during the data collection periods and who were ?discharged from the ED by the end of the data collection period?

Why were patients with minor injuries excluded?

Was a data collection sheet used? If so, could you provide it?

Was the date of PIVC insertion recorded?

Were the indications for PIVC used recorded? This isn't actually stated in the methods. Were pre-determined categories used?

Were the diagnoses/likely diagnoses for the included patients recorded?

What was the length of stay for the included patients?

IV medications is listed as one of the indications for giving IV fluids- Have you classified IV medication as IV fluids? Please clarify

Authors state that patients were included if in a non-resus area and excluded if transferred to a resus area- fluid resuscitation is listed as one of the indications for IV fluids- seems to clash with the inclusion/exclusion criteria- authors please clarify

How did the team determine the indications for the administration of fluids? Did they ask the clinicians managing the patient or check the patient notes? (noted that data was collected using patient observations, records and clinicians but it would be good to specifically say how the indications for use were determined)

Definitions of unnecessary and necessary use of PIVC should be presented earlier

Results

Definitions for the categories should be presented in the methods not the results – the ‘unused and necessary’ category was not previously mentioned or defined

Would ‘unused’ PIVC also be considered ‘unnecessary’?

Table 2- were these categories pre-determined? It is not uncommon in an ED setting to place an IV in the event the patient will be admitted or deteriorates- was that indication included in the data collection?

Discussion- this could be more concise, The numerous subheadings are not necessary for this study. Focus on what you found and possible explanations for the findings. The ‘just in case’ category was not noted as in indication in the current study- is it that there were no ‘just in case’ scenarios or was this not collected?

The study was conducted in two countries- someone discussion around similarities/differences would be useful.

Reviewer #2: This paper requires edits prior to publication. Firstly, it is not the largest study to examine the utilisation of PIVC within the ED setting: Hawkins T et al. Peripheral Intravenous Cannula Insertion and Use in the Emergency Department: An Intervention Study. Acad Emerg Med. 2018 Jan;25(1):26-32. Secondly, an explanation as to why the two sites were chosen as comparison or similar sites is required. More detail into the methods as to why differing time frames, sampling methods and data collection processes were used is needed to strengthen the study. The differences are a significant risk to the methods and need to be explained. Additional explanation as to the data analysis methods used would be of benefit including reworking the tables as in the current format the results are unclear.

7. PLOS authors have the option to publish the peer review history of their article (what does this mean?). If published, this will include your full peer review and any attached files.

Reviewer #1: No

Reviewer #2: No

---

## [Author Response · Author response to Decision Letter 1]

14 Jan 2024

Please see the full responses to reviewers in the letter titled "Response to reviewers". We have also copied those responses below:

Reviewer #1

Introduction

• Suggest re-structuring the introduction- 4th paragraph can be integrated with the first, then follow with risks associated with PIVC and then discuss unused/unnecessary PIVC. 

o With thanks. We have restructured the paragraph as suggested

• Authors could also consider discussing the environmental impact of the inappropriate use of PIVC

o We could not find studies comparing simple phlebotomy vs PIVC insertion and the environmental impact of this. Therefore we have added a line (76-78) to reflect this important point.

• Line 63: where were the other studies conducted?

o Webster et al. 2019 is a Cochrane review which included 7 trials. 6 were single centre, and one was a multi-centre trial. One trial was based in India, one in the UK, and the rest in Australia. 6 were based in acute inpatient settings, and one in a community hospital setting. Maki et al. 2016 included 20 prospective studies in their systematic review observing blood stream infection from PIVCs. 7 studies were based on medical inpatient wards, 7 on surgical wards, 2 in ITU and 4 in haematology/oncology wards. We have added a line (70-71) to reflect this.

• Lines 63-65- the sentence related to OR is not quite clear: are you saying that the risk of catheter associated infections is 6 times greater when inserted in the ED compared to inpatient wards? What is the p value? I’m not sure what the OR of 2.3 is referring to. Please also include where this study was conducted and if it was a large or small study eg. Are these results from a single centre ED in X country?

o The study Trinhh et al. 2011 was a single tertiary centre hospital in the US. It was a retrospective case note review over a 3 year period. The authors identified a total of 544 staphylococcus aureus bacteraemias, of which 24 were identified to be caused by the PIVC through clinical signs (phlebitis around the site) and positive tip cultures of the device. 16 of the PIVC-related S. aureus bacteremias (67%), the PIVC was placed in the emergency department, 4 (17%) were placed in an inpatient unit, 2 (8%) were placed by emergency medical services prior to admission, and 2 (8%) were placed at outside hospitals. The risk is 6 times greater for cannulas inserted in the ED (OR 6.03, p< 0.001) compared with those inserted as an inpatient (0.17, p<0.001). Note the previous reported OR of 0.23 was a mistake. The total were compared with the total PIVCs observed prospectively over a one day period (317 total PIVCs in situ with 79 inserted in A&E and 170 inserted on the ward). Note the confidence intervals are reported incorrectly for the OR of PIVCs inserted in the emergency department, with a typo at the upper bound. We could not find a revision submitted by the authors. The Fischer’s exact test is <0.001 for both. I have included the table from the paper for reference.

o We have updated the introduction to reflect these points in lines 72-73

• What is the rational for focusing only on the use of PIVC for IV fluids? What about unnecessary use related to medication? Or generally on the appropriate use of PIVC?

o There is previous evidence to show IV fluids were being utilised for inappropriate indications in the emergency department, examples include in renal colic and alcohol intoxication where IV fluids are frequently prescribed despite data showing no changes in outcomes. They are also the most common intervention given via a PIVC secondary to usage for phlebotomy. We felt it was more appropriate to focus on inappropriate IV fluid usage as we could formulate a priori indications related to known Trust guidelines. Given the heterogeneity and number of different medications given in a tertiary ED we felt it would be difficult to introduce a priori indications for this category. It is also hard to define appropriate vs inappropriate medications around delivering these as IV or PO. We could use guidelines but there are many good reasons the treating team may have deviated from these which were not apparent to the researchers. Furthermore the main data collectors were medical students in their 3rd-4th year of study, who would not be able to clinically determine every medication’s indication and if this was suitably given without a priori indications. 

• The first part of the aim isn’t quite clear- are you assessing the prevalence of PIVC use in the ED or are you describing the indications for PIVC use in the ED?

o The first aim is to describe both the prevalence and their usage. This has been clarified in lines 80-81. The secondary aim was to assess the indications for IV fluid usage, and categorise PIVCs in to used, unused, and unnecessary.

Methods

• Need to specifically say that the study site were ED centers

o This is described in the introduction in lines 80-81, we have also added a clarification point in line 87 in the methods.

• Suggest providing a brief description of the ED sites- size, volume of patients, staffing, training, types of patients

o We have updated the methodology to reflect this in lines 87-90

• Selection of participants might be better presented as inclusion and exclusion criteria. Unlikely that all patients would have been followed given the design used. Rather the inclusion criteria would have been adult patients, presenting to the ED with X conditions during the data collection periods and who were ?discharged from the ED by the end of the data collection period?

o This has been clarified in lines 102-106. 

• Why were patients with minor injuries excluded?

o The treatment of patients in minor injuries typically did not involve inserting a PIVC. If patients required IV medications or fluid, they were usually transferred into majors and would have been captured by our inclusion criteria with a PIVC generally inserted in majors.

• Was a data collection sheet used? If so, could you provide it?

o Data was collected using an electronic tablet on an online spreadsheet. The categories recorded are included in lines 116-117.

• Was the date of PIVC insertion recorded?

o Yes, this is detailed in lines 119-123. The date and time of PIVC insertion, and date and time of patient discharge was recorded. 

• Were the indications for PIVC used recorded? This isn't actually stated in the methods. Were pre-determined categories used?

o The only indications were the a priori indications which are listed in lines 125-129. We did not otherwise record PIVC indications.

• Were the diagnoses/likely diagnoses for the included patients recorded?

o No, only the a priori indications for IV fluid usage.

• What was the length of stay for the included patients?

o We have not included this in our analysis, however it would have been a maximum of 14 hours in Centre A or 6 hours in Centre B (if patients stayed longer they would have been excluded as part of our study recruitment exclusions)

• IV medications is listed as one of the indications for giving IV fluids- Have you classified IV medication as IV fluids? Please clarify

o Some IV medications/contrast/blood products require an IV “push” of fluid. This was included in the a priori exclusions as the IV fluid is needed as part of administering medications. We have clarified this point in the methodology lines 126-127.

• Authors state that patients were included if in a non-resus area and excluded if transferred to a resus area- fluid resuscitation is listed as one of the indications for IV fluids- seems to clash with the inclusion/exclusion criteria- authors please clarify

o Some patients in the “majors” area were given IV fluid resuscitation if the resus area was full, or were not critically unwell enough to warrant transfer as decided by a senior clinician. This point captured those patients.

• How did the team determine the indications for the administration of fluids? Did they ask the clinicians managing the patient or check the patient notes? (noted that data was collected using patient observations, records and clinicians but it would be good to specifically say how the indications for use were determined)

o As noted in lines 117-119 “Data concerning PIVC insertion and utilisation were prospectively recorded from patient arrival until discharge from the ED through direct observation, prospective ED record review and/or clarification with the patient’s clinician”

o The specific indications are as described in the “a priori” indication list. 

• Definitions of unnecessary and necessary use of PIVC should be presented earlier

o This has been moved to earlier in the methodology section

Results

• Definitions for the categories should be presented in the methods not the results – the ‘unused and necessary’ category was not previously mentioned or defined

o Definitions are listed in the methodology and we have included a definition for “unused” PIVCs, and we have felt that repeating the definitions in the results table provides clarity to those who may not read the full paper.

• Would ‘unused’ PIVC also be considered ‘unnecessary’?

o Yes, but we wanted to distinguish those unnecessary PIVCs that were utilised inappropriately for IV fluid administration, therefore we have used both unused and unnecessary

• Table 2- were these categories pre-determined? It is not uncommon in an ED setting to place an IV in the event the patient will be admitted or deteriorates- was that indication included in the data collection?

o We acknowledge the argument for ‘just in case PIVCs’. However predicting clinical risk of deterioration is challenging and risks many PIVCs being placed per use and may be regarded as wasteful. We have noted patients who had a PIVC inserted, and who subsequently deteriorated requiring admission to a resuscitation area as very low (see discussion). We did not recruit patients in the resuscitation area who are at higher risk in part for this reason. Liberal placement of PIVCs may also promote inappropriate use of IV fluids. In addition PIVCs take longer to insert, and have a higher environmental impact. There was otherwise no indication in the data collection for “just in case”

Discussion

• This could be more concise, The numerous subheadings are not necessary for this study. Focus on what you found and possible explanations for the findings. The ‘just in case’ category was not noted as in indication in the current study- is it that there were no ‘just in case’ scenarios or was this not collected?

o We have rewritten the discussion to be more concise, removed sections that do not explicitly relate to our results, and removed the subheadings. We would appreciate the reviewers’ opinion on removing the subheadings, as we feel this has now reduced some of the structure and clarity in the discussion.

o The “just in case” category was not included as an indication (see above answer). We have demonstrated that 11 patients with a PIVC were transferred to resus, and utilised this information to inform further discussion around the need for “just in case” PIVCs

• The study was conducted in two countries- someone discussion around similarities/differences would be useful.

o We discuss the similarities in the methodology section, but have chosen not to analyse the population differences between the two centres as this study was not powered or designed to be a comparison between two different hospitals. We have added lines in the discussion around some of the differences (159-163)

Reviewer #2: 

• This paper requires edits prior to publication. Firstly, it is not the largest study to examine the utilisation of PIVC within the ED setting: Hawkins T et al. Peripheral Intravenous Cannula Insertion and Use in the Emergency Department: An Intervention Study. Acad Emerg Med. 2018 Jan;25(1):26-32. 

o This is the largest international prospective study to date. It was not single centre, and we wanted to reflect this. However we can understand this line may be misleading, and so we have edited this in lines 153-154

• Secondly, an explanation as to why the two sites were chosen as comparison or similar sites is required. More detail into the methods as to why differing time frames, sampling methods and data collection processes were used is needed to strengthen the study. The differences are a significant risk to the methods and need to be explained. 

o These are well made points. The data collection primarily reflected the availability of the research team. The study was not funded and all researchers completed training and data collection in their own time which was limited by their own work and study. The two sites were chosen as there were established academic links between them and existing research infrastructure, so making roll out of the study more likely to succeed. There is explanation in the limitations (lines 236-237) that describe the reasons for the differing time frames and sampling methods were used – due to data collector availability. We are happy to be advised if the reviewer suggests additional or alternative wording.

o We have added a section around the main differences in the hospitals in the discussion.

• Additional explanation as to the data analysis methods used would be of benefit including reworking the tables as in the current format the results are unclear.

o Thank you. We have reformatted the tables to provide more clarity with the results and more consistency with matching definitions.

---

## [Decision Letter · Decision Letter 2]

26 Mar 2024

PONE-D-23-09905R2Prospective observational study of peripheral intravenous cannula utilisation and frequency of intravenous fluid delivery in the emergency department – convenience or necessity?PLOS ONE

Dear Dr. Willis,

Thank you for submitting your manuscript to PLOS ONE. After careful consideration, we feel that it has merit but does not fully meet PLOS ONE’s publication criteria as it currently stands. Therefore, we invite you to submit a revised version of the manuscript that addresses the points raised during the review process.

Please revise.

We look forward to receiving your revised manuscript.

Kind regards,

Academic Editor

PLOS ONE

Journal Requirements:

Reviewers' comments:

Reviewer's Responses to Questions

**Comments to the Author**

1. If the authors have adequately addressed your comments raised in a previous round of review and you feel that this manuscript is now acceptable for publication, you may indicate that here to bypass the “Comments to the Author” section, enter your conflict of interest statement in the “Confidential to Editor” section, and submit your "Accept" recommendation.

Reviewer #1: All comments have been addressed

Reviewer #2: (No Response)

2. Is the manuscript technically sound, and do the data support the conclusions?

Reviewer #1: Yes

Reviewer #2: Yes

3. Has the statistical analysis been performed appropriately and rigorously? 

Reviewer #1: N/A

Reviewer #2: Yes

4. Have the authors made all data underlying the findings in their manuscript fully available?

Reviewer #1: Yes

Reviewer #2: Yes

5. Is the manuscript presented in an intelligible fashion and written in standard English?

Reviewer #1: Yes

Reviewer #2: Yes

6. Review Comments to the Author

Reviewer #1: Thanks for addressing the comments.

The only minor comment is if the authors could include how the criteria for administration of IVF was determined. Is it based on existing guidance or was it developed by the authors/expert team for the study.

Reviewer #2: Thank you for your edits to date. I would suggest the following minor edits prior to publication- Line 71- please expand ITU to explain the term. Line 159- should be 'centre' not 'centres'. Line 175- your results conflict with this study, in which the authors accept PIVC utilisation for IV fluids as an acceptable reason for PIVC insertion. I suggest you link this to your paragraph (line 191) which explains this further. In the current format it does not flow (ie move the paragraph involving lines 179-189 down further to line 207).

7. PLOS authors have the option to publish the peer review history of their article (what does this mean?). If published, this will include your full peer review and any attached files.

Reviewer #1: No

Reviewer #2: No

---

## [Author Response · Author response to Decision Letter 2]

1 May 2024

1 May 2024

Dear Dr Chen,

We thank you and the reviewers for the considered comments and review of the original research article entitled “Prospective observational study of peripheral intravenous cannula utilisation and frequency of intravenous fluid delivery in the emergency department – convenience or necessity?” 

Please see below our responses to the reviewers comments, please note the line references in our responses are for the document entitled “Manuscript”:

Reviewer #1

• Thanks for addressing the comments. The only minor comment is if the authors could include how the criteria for administration of IVF was determined. Is it based on existing guidance or was it developed by the authors/expert team for the study

o These criteria were based on existing NICE (National Institute for Health and Clinical Excellence) guidance on indications for IV fluid in adults, we have clarified and referenced this in lines 127-127.

Reviewer #2

• Thank you for your edits to date. I would suggest the following minor edits prior to publication- Line 71- please expand ITU to explain the term. 

o We have adjusted “ITU” to “ICU” Intensive Care Unit as this is a more internationally recognised term in line 71

• Line 159- should be 'centre' not 'centres'. 

o This has been edited.

• Line 175- your results conflict with this study, in which the authors accept PIVC utilisation for IV fluids as an acceptable reason for PIVC insertion. I suggest you link this to your paragraph (line 191) which explains this further. 

o We have moved this section on Hawkins et al. 2017 to lines 188-192, linking the distinction our work has made to identifying IV fluid treatment with a priori criteria.

• In the current format it does not flow (ie move the paragraph involving lines 179-189 down further to line 207).

o We agree and have moved the paragraph at your suggestion.

Thank you once again for your time in considering this work.

Yours sincerely,

Dr Michael Willis

BSc, MBBS, MRCPCH

---

## [Decision Letter · Decision Letter 3]

28 May 2024

Prospective observational study of peripheral intravenous cannula utilisation and frequency of intravenous fluid delivery in the emergency department – convenience or necessity?

PONE-D-23-09905R3

Dear Dr. Willis,

We’re pleased to inform you that your manuscript has been judged scientifically suitable for publication and will be formally accepted for publication once it meets all outstanding technical requirements.

Kind regards,

Academic Editor

PLOS ONE

Additional Editor Comments (optional):

Reviewers' comments:

Reviewer's Responses to Questions

**Comments to the Author**

1. If the authors have adequately addressed your comments raised in a previous round of review and you feel that this manuscript is now acceptable for publication, you may indicate that here to bypass the “Comments to the Author” section, enter your conflict of interest statement in the “Confidential to Editor” section, and submit your "Accept" recommendation.

Reviewer #1: All comments have been addressed

Reviewer #2: All comments have been addressed

2. Is the manuscript technically sound, and do the data support the conclusions?

Reviewer #1: Yes

Reviewer #2: Yes

3. Has the statistical analysis been performed appropriately and rigorously? 

Reviewer #1: N/A

Reviewer #2: Yes

4. Have the authors made all data underlying the findings in their manuscript fully available?

Reviewer #1: Yes

Reviewer #2: Yes

5. Is the manuscript presented in an intelligible fashion and written in standard English?

Reviewer #1: Yes

Reviewer #2: Yes

6. Review Comments to the Author

Reviewer #1: Thank you to the authors for addressing the comments.

I have no further comments for this manuscript.

Look forward to the final product

Reviewer #2: The authors have addressed comments and concerns. The manuscript is now appropriate for publication.

7. PLOS authors have the option to publish the peer review history of their article (what does this mean?). If published, this will include your full peer review and any attached files.

Reviewer #1: No

Reviewer #2: No
